

# Does $\delta^{18}O$ of $O_2$ record meridional shifts in tropical rainfall?

Alan M. Seltzer,[1] Christo Buizert[2], Daniel Baggenstos[3], Edward J. Brook[2], Jinho Ahn[4], Ji-Woong Yang[4], Jeffrey P. Severinghaus[1]

[1]Scripps Institution of Oceanography, University of California-San Diego, La Jolla, CA, 92037, USA
[2]College of Earth, Ocean and Atmospheric Sciences, Oregon State University, Corvallis, OR, 97331, USA
[3]Climate and Environmental Physics, University of Bern, Bern, 3012, Switzerland
[4]School of Earth and Environmental Sciences, Seoul National University, Seoul, 08826, South Korea

*Correspondence to*: Alan M. Seltzer (aseltzer@ucsd.edu)

**Abstract.** Marine sediments, speleothems, paleo lake elevations, and ice core methane and $\delta^{18}O$ of $O_2$ ($\delta^{18}O_{atm}$) records provide ample evidence for repeated abrupt meridional shifts in tropical rainfall belts throughout the last glacial cycle. To improve understanding of the impact of abrupt events on the global terrestrial biosphere, we present composite records of $\delta^{18}O_{atm}$ and inferred changes in fractionation by the global terrestrial biosphere ($\Delta\varepsilon_{LAND}$) from discrete gas measurements in the WAIS Divide (WD) and Siple Dome (SD) Antarctic ice cores. On the common WD timescale, it is evident that maxima in $\Delta\varepsilon_{LAND}$ are synchronous with or shortly follow WD $CH_4$ peaks assumed to mark abrupt climate responses to Heinrich events. Based on our analysis of the modern seasonal cycle of gross primary productivity (GPP)-weighted $\delta^{18}O$ of terrestrial precipitation (the source water for atmospheric $O_2$ production), we propose a simple mechanism by which $\Delta\varepsilon_{LAND}$ tracks the centroid latitude of terrestrial oxygen production. As intense rainfall and oxygen production migrate northward, $\Delta\varepsilon_{LAND}$ should decrease due to the underlying meridional gradient in rainfall $\delta^{18}O$. A southward shift should increase $\Delta\varepsilon_{LAND}$. Monsoon intensity also influences $\delta^{18}O$ of precipitation, and although we cannot determine the relative contributions of the two mechanisms, both act in the same direction. Therefore, we suggest that abrupt increases in $\Delta\varepsilon_{LAND}$ unambiguously imply a southward shift of tropical rainfall. The exact magnitude of this shift, however, remains under-constrained by $\Delta\varepsilon_{LAND}$.

## 1 Introduction

The last glacial cycle shows two modes of abrupt climate variability, both with a center of action in the North Atlantic. The first are Dansgaard-Oeschger (D-O) events, most clearly identified in Greenland ice core records (Andersen et al., 2004; Dansgaard et al., 1993) most of which can be linked to abrupt variations in the strength and/or northward heat transport of the Atlantic Meridional Overturning Circulation (Lynch-Stieglitz, 2017). The second are Heinrich events (HE), periods of extreme cold in the North-Atlantic associated with extensive layers of ice rafted detritus in ocean. Heinrich events occur within longer Heinrich stadial (HS) periods (Hemming, 2004).

High-resolution records of marine sediments (Peterson et al., 2000), ice core methane (Rhodes et al., 2015), speleothem calcite $\delta^{18}$O (Cheng et al., 2016; Kanner et al., 2012; Wang et al., 2017, 2008, 2001), and lake elevations (Goldsmith et al., 2017; Yu et al., 2013) all demonstrate that abrupt meridional shifts in tropical rainfall belts occurred repeatedly during the last glacial cycle in response to both D-O events and HEs. This response is also observed in climate models, which consistently

simulate a shift of tropical rainfall towards the warmer hemisphere (Broccoli et al., 2006; Chiang and Bitz, 2005; Cvijanovic et al., 2013). Understanding the impact of these events on global hydrology, monsoon systems, and terrestrial biosphere remains a key goal of paleoclimate research.

Measurements of long-lived atmospheric gases trapped in ice cores offer an opportunity to better constrain past global processes due to the short atmospheric mixing time and therefore globally integrated signal of these gases. Measurements of

concentrations and isotopic ratios of atmospheric gases have provided important clues into the biogeochemical responses to substantial climate changes in the past. Motivated by the recent high-resolution West Antarctic Ice Sheet (WAIS) Divide (WD) ice core methane record (Rhodes et al., 2015), in this study we explore a new record of atmospheric oxygen isotopes from WD and revisit published measurements in the Siple Dome (SD) ice core. A composite WD-SD record of atmospheric oxygen isotopes allows us to investigate past changes in the isotopic fractionation of $O_2$ on a common timescale (Buizert et al., 2015)

with other recent records.

Past changes in the Dole Effect – the amount by which $\delta^{18}$O of atmospheric $O_2$ ($\delta^{18}O_{atm}$) exceeds that of seawater (Dole, 1935; Morita, 1935) – have been shown to be closely linked to terrestrial hydroclimate (Bender et al., 1994; Landais et al., 2010; Reutenauer et al., 2015; Severinghaus et al., 2009). We quantitatively explore a simple relationship between meridional shifts in terrestrial oxygen production and $\delta^{18}$O of oxygen production-weighted terrestrial rainfall ($\delta^{18}O_{precip}$) over the modern

seasonal cycle using spatially gridded monthly observations. We use these observations as a guide to consider instantaneous changes in $O_2$ fractionation during HEs and discuss the implications of these changes, and related changes in atmospheric $CH_4$, for the meridional distribution of tropical rainfall associated with these events.

## 1.1 The Siple Dome and WAIS Divide ice cores

The 1004-meter-long SD ice core (81.65° S 148.81° W) was drilled in the late 1990s at an elevation of 621 m above sea

level. The site has modern-day mean-annual surface temperature and accumulation rates of -24.5°C and 13.5 cm ice equivalent



per year, respectively (Taylor et al., 2004). The youngest 8.2 ka of the record were dated by visual and electrical counting of annual layers (Taylor et al., 2004), tuned to stratigraphic markers, while the older portion of the ice core was dated by synchronization of $CH_4$ and $\delta^{18}O_{atm}$ measurements to the GISP2, Greenland ice core (Brook et al., 2005). The 3,404-meter-long WD ice core (79.48° S, 112.11° W) was drilled at an elevation of 1766 m between 2007 and 2011. Present mean-annual

air temperature and accumulation rate at the WD ice core site are -30°C and 22 cm ice equivalent per year (Banta et al., 2008; Morse et al., 2002).The youngest 31.2 ka (2850 m) was dated by annual-layer counting (Sigl et al., 2016) while the oldest portion of the core (up to ~68 ka) was dated by $CH_4$ synchronization to the Greenland NGRIP ice core, scaled linearly to match the timing of abrupt events recorded in U/Th dated speleothem records (Buizert et al., 2015).

### 1.2 $\delta^{18}O_{atm}$ and the Dole Effect: a brief overview

The enrichment of $\delta^{18}O_{atm}$ relative to $\delta^{18}O$ of mean seawater ($\delta^{18}O_{sw}$) is known as the Dole Effect (Dole, 1935; Morita, 1935) and has a modern-day value of ~23.88‰ (Barkan and Luz, 2005). Most of the Dole Effect is attributable to the isotopic discrimination of marine and terrestrial respiration, both of which preferentially utilize $^{16}O$, thus enriching atmospheric $O_2$ in $^{18}O$ (Bender et al., 1994; Luz and Barkan, 2011). Photosynthesis does not fractionate the $^{18}O/^{16}O$ ratio of chloroplast water (e.g. Helman et al., 2005), and therefore the $\delta^{18}O$ of $O_2$ produced by photosynthesis is equal to the $\delta^{18}O$ of the plant source

water (ultimately derived from precipitation) plus fractionation due to evapotranspiration. Past changes in the Dole Effect estimated from measurements of atmospheric oxygen isotopes in ice core bubbles and reconstructions of $\delta^{18}O_{sw}$ from marine sediment cores have been shown to be small over the past 130 ka, during which the standard deviation of the Dole Effect was only 0.24‰ (Bender et al., 1994).

      Millennial-scale variability in the Dole Effect has been hypothesized to be related to terrestrial hydrology based on

coherence with atmospheric $CH_4$ measured in ice core bubbles (Bender et al., 1994), comparison with Chinese cave records (Severinghaus et al., 2009) and climate modeling experiments (Reutenauer et al., 2015). To investigate instantaneous changes in the isotopic fractionation of atmospheric $O_2$, *Severinghaus et al.* (2009) used a one-box model deconvolution to account for the smoothing effect of the ~1000 year residence time of atmospheric $O_2$. The parameter $\Delta\varepsilon_{LAND}$, associated with this box model, approximately represents changes from the present in globally integrated (mostly) terrestrial $O_2$ isotopic fractionation:

$$\Delta\varepsilon_{LAND} \equiv \Delta\varepsilon_L - \Delta\varepsilon_{RL} - \Delta\varepsilon_{RO}\left(1 - \frac{1}{f_L}\right) \qquad (1)$$

where $\Delta\varepsilon_L$ refers to the change from present in the fractionation of terrestrially produced oxygen relative to seawater, $\Delta\varepsilon_{RL}$ and $\Delta\varepsilon_{RO}$ respectively refer to changes from present in terrestrial and marine effective respiratory fractionation, and $f_L$ is the fraction of oxygenesis occurring on land. Because the three terms in equation 1 are not independently discernible, *Severinghaus et al.* (2009) derive a formula for $\Delta\varepsilon_{LAND}$ in terms of knowable quantities: $\delta^{18}O_{sw}$, $\delta^{18}O_{atm}$ and time derivative of $\delta^{18}O_{atm}$

($d\delta^{18}O_{atm}/dt$). The calculation of $\Delta\varepsilon_{LAND}$ in terms of these variables is described in section 2.1. $\Delta\varepsilon_{LAND}$, as estimated from SD $\delta^{18}O_{atm}$ measurements, is strongly correlated with Dongge cave $\delta^{18}O$ (Wang et al., 2005, 2001) over the past 12 ka (Severinghaus et al., 2009). Before this period, uncertainty in the original SD gas timescale prevented reliable high-precision comparison with other paleoclimate records. Improving the timescale for SD $\delta^{18}O_{atm}$ measurements is therefore a key goal of this work.

Modeling results from a recent freshwater hosing experiment suggest that changes in $\delta^{18}O_{atm}$ over a HS are dominated by changes in $\delta^{18}O_{precip}$ over the terrestrial biosphere (Reutenauer et al., 2015). In terms of equation 1, this finding suggests that $\Delta\varepsilon_{LAND}$ may be driven by $\Delta\varepsilon_L$. Below we build on the conclusion of this recent modeling study by exploring a composite WD-SD $\Delta\varepsilon_{LAND}$ record on an improved timescale and drawing insights from the modern seasonal cycles of terrestrial oxygenesis and $\delta^{18}O_{precip}$.

## 2 Methods

### 2.1 $\delta^{18}O_{atm}$ measurements, curve fitting and determination of $\Delta\varepsilon_{LAND}$

Ice samples from WAIS Divide were analyzed at Scripps Institution of Oceanography (SIO) between 2009 and 2016. A total of 1,037 ~15g ice samples were measured for $\delta^{15}N$ of $N_2$, $\delta^{18}O_{atm}$, $\delta O_2/N_2$ and $\delta Ar/N_2$, following the procedures of *Petrenko et al.* (2006) and *Severinghaus et al.* (2009). Measured values are reported relative to ratios measured in atmospheric

air collected off Scripps Pier (La Jolla, CA). 560 unique depths were sampled, with 434 measured multiple times (2-4 replicate samples). To determine $\delta^{18}O_{atm}$, measured $\delta^{18}O$ of $O_2$ was corrected for gravitational settling by $\delta^{15}N$ and for gas loss by pair differences of $\delta O_2/N_2$ and $\delta Ar/N_2$, following *Severinghaus et al.* (2009). A brief overview of these corrections is given in Appendix A. The pooled standard deviation of replicate gravity-and-gas-loss corrected $\delta^{18}O$ measurements at WD is 0.0085‰.



In order to evaluate a smooth derivative for the calculation of $\Delta\varepsilon_{LAND}$, discrete WD $\delta^{18}O_{atm}$ values were fit to a Fourier series ranging from 1/697 to 3.2 cycles ka$^{-1}$ (spacing: 1/697 cycles ka$^{-1}$) by a Bayesian, weighted, linear least-squares technique, which *a priori* assumes a red spectrum and factors this assumption into the cost function (Severinghaus et al., 2009, Appendix A). The time derivatives of SD (on its original timescale) and WD $\delta^{18}O_{atm}$ fitted curves were used to determine depth-gas age tie points between the two cores.

After synchronization of SD gas ages to the WD2014 timescale (Section 2.2), a composite fitted curve and time derivative of $\delta^{18}O_{atm}$ were calculated using all discrete SD and WD $\delta^{18}O_{atm}$ younger than 50.12 ka BP. $\Delta\varepsilon_{LAND}$ was calculated independently in three ways, from fitted $\delta^{18}O_{atm}$ curves and derivatives in 1) SD only, 2) WD only, and 3) the composite SD-WD record, each time using the same smoothed, interpolated $\delta^{18}O_{sw}$ record (Waelbroeck et al., 2002):

$$\Delta\varepsilon_{LAND} = \frac{1}{f_L}(\delta^{18}O_{atm} - \delta^{18}O_{sw} + \tau \frac{d\delta^{18}O_{atm}}{dt}) \tag{2}$$

where $f_L$ is the fraction of photosynthesis occurring on land (assumed to be 0.65) and $\tau$ is the residence time of atmospheric $O_2$ (1 ka).

**2.2 Synchronization of Siple Dome gas ages to WAIS Divide**

Siple Dome gas ages between 50 ka and 1950 CE were synchronized with the WAIS Divide WD2014 timescale (Buizert et al., 2015) using the smooth annual layer thickness (ALT) method of *Fudge et al.* (2014). The ALT method, as applied in this study, is a standard regularized linear least-squares method (Aster et al., 2013) that estimates an annual layer thickness history constrained by depth-gas age tie points and smoothness of the ALT time series. Integration of the resulting time series of annual layer thicknesses yields a gas age-depth chronology. Formally, the ALT method seeks an optimal annual layer thickness history, **m**, that minimizes the cost function C as defined:

$$C = \left\| \frac{\mathbf{Gm\text{-}d}}{\sigma} \right\|_2^2 + \alpha \|\mathbf{Lm}\|_2^2 \tag{3}$$

where **G** is a matrix that maps annual layer thicknesses **m** to depths **d** corresponding to tie points of a given gas age. **L** is a second-derivative operator and $\alpha$ is a trade-off parameter between smoothness and tie-point agreement (the notation $\| \ \|_2^2$



indicates the squared L2 norm of the argument). Depth constraints are weighted based on individual estimates of tie point uncertainty, $\boldsymbol{\sigma}$, as discussed in detail in Appendix B.

In total, 56 tie points were used to constrain the ALT inverse problem: 36 from SD and WD records of $d\delta^{18}O_{atm}/dt$, and

20 from SD and WD $CH_4$ records. Because $\delta^{18}O_{atm}$ is smoothly varying on millennial timescales due to the ~1 ka atmospheric residence time of $O_2$, WD and SD $d\delta^{18}O_{atm}/dt$ time series (calculated from fitted curves of $\delta^{18}O_{atm}$ at 10-year resolution, Appendix A) were instead used for synchronization. Tie points based on $d\delta^{18}O_{atm}/dt$ were determined by calculating the midpoint times of abrupt transitions common to both records. $CH_4$-based tie points were determined by value-matching unambiguous abrupt transitions identified in the two records, within a prescribed error range of $\pm 15$ ppb. More complete explanations of the tie-point selection process, an estimate of its uncertainty, the application of the ALT method, and sensitivity

tests are provided in Appendix B. Figure 1 shows the fitted SD $\delta^{18}O_{atm}$ curve on its original (Brook et al., 2005) and WD2014-synchronized timescales alongside the WD $\delta^{18}O_{atm}$ discrete measurements and fitted curve. Figure 2 shows the synchronization results as a comparison between the original and new SD gas timescales, $CH_4$ and $d\delta^{18}O_{atm}/dt$ tie points, and the annual (gas) layer thickness profile of the new and original timescales.

## 2.3 Modern seasonal cycle analysis

To explore seasonal changes in the precipitation supplied to terrestrial oxygen production (Section 3.2), mean monthly gridded datasets of gross primary productivity (GPP; Jung et al., 2011) and $\delta^{18}O_{precip}$ (Terzer et al., 2013) were analyzed. The $\delta^{18}O_{precip}$ isoscape product used for our analysis (Section 3.2) was created from regionalized cluster-based water isotope prediction (RCWIP) constrained by 57,000 Global Network of Isotopes in Precipiation (GNIP) measurements from 1960 to 2009 (IAEA, 2017; IAEA/WMO, 2017; Terzer et al., 2013). The gridded monthly mean GPP product used in our study was

determined from upscaled eddy covariance observations (FLUXNET) from 1982-2011 (Jung et al., 2011).

To estimate monthly terrestrial GPP-weighted $\delta^{18}O_{precip}$, monthly-mean isoscapes (originally 1/6° x 1/6°) were regridded using MATLAB's geoloc2grid function to 0.5° x 0.5° resolution to match the resolution of the GPP dataset. GPP data were converted from carbon (or, equivalently, oxygen after accounting for a photosynthetic quotient) fluxes to total emissions by multiplying by grid cell area. In equations 4 and 5 below, GPP refers to an emission, rather than a flux, and is therefore

independent of latitude-varying grid cell area. Only land area between 60°S and 84°N was included due to data availability,



with several gaps over low-GPP regions like the Sahara Desert and Greenland. Monthly global terrestrial GPP-weighted mean $\delta^{18}O_{precip}$ was calculated as follows:

$$\overline{\delta^{18}O_{precip}}(t) = \frac{\int_{-60}^{84}\int_{0}^{360} GPP(\varphi,\lambda,t)\delta^{18}O_{precip}(\varphi,\lambda,t)d\varphi d\lambda}{\int_{-60}^{84}\int_{0}^{360} GPP(\varphi,\lambda,t)d\varphi d\lambda} \tag{4}$$

where $\overline{\delta^{18}O_{precip}}(t)$ is terrestrial GPP-weighted mean $\delta^{18}O_{precip}$ during month $t$, and $\varphi$ and $\lambda$ refer to center latitudes and

longitudes of terrestrial grid cells with spacing $d\lambda = d\varphi = 0.5°$.

The centroid latitude of terrestrial oxygenesis ($\varphi_{TOE}$) is defined as the latitude at which $f_{GPP}$, the meridionally cumulative fraction total monthly GPP, equals 0.5. Integrating northward from 60°S, we define $\varphi_{TOE}$ such that:

$$f_{GPP}(\varphi_{TOE}) = \frac{\int_{-60}^{\varphi_{TOE}}\int_{0}^{360} GPP(\varphi,\lambda,t)d\lambda d\varphi}{\int_{-60}^{84}\int_{0}^{360} GPP(\varphi,\lambda,t)d\lambda d\varphi} = 0.5 \tag{5}$$

In practice, we estimate $\varphi_{TOE}$ at finer resolution than the 0.5° spacing by first calculating zonal integrals around each band of

grid cells and then linearly interpolating such that $d\varphi$ in equation 5 equals 0.006.

## 3 Results

### 3.1 $\Delta\varepsilon_{LAND}$ variations over past 50 ka

The WD-SD composite $\Delta\varepsilon_{LAND}$ record is shown in Figure 3 alongside a composite East Asian speleothem record of calcite $\delta^{18}O$ (Cheng et al., 2016) and discrete and continuous $CH_4$ records from WD (Rhodes et al., 2015; Sowers, 2012). The

speleothem $\delta^{18}O$ record has been corrected for seawater $\delta^{18}O$ changes (Waelbroeck et al., 2002) and is shown both unsmoothed and smoothed (200-yr boxcar filter). There are several notable features of the composite $\Delta\varepsilon_{LAND}$ time series and its comparison to these other records.

First, $\Delta\varepsilon_{LAND}$ is positively and significantly correlated with the seawater-corrected East Asian speleothem $\delta^{18}O$ record (r = 0.70, p < 0.0001, both records interpolated linearly to 200-year resolution between 50 and 0 ka BP). The regression

coefficient, β, over the entire 50-ka time window is 0.215 ‰$_{atm}$ ‰$_{pdb}^{-1}$. β varies between 0.387 ‰$_{atm}$ ‰$_{pdb}^{-1}$ during the Holocene (0-12 ka) to 0.168 ‰$_{atm}$ ‰$_{pdb}^{-1}$ during the last glacial period (12-50 ka). Although the reason for the glacial-interglacial change in β is unclear and a worthy topic of future study, the high correlation (r = 0.62, p < 0.0001) beyond the ~0-12 ka window originally considered by *Severinghaus et al.* (2009) confirms that the $\Delta\varepsilon_{LAND}$ and the Asian cave records share common



variability throughout the last glacial period in addition to the Holocene. We note that the misalignment of $\Delta\varepsilon_{LAND}$ and Chinese cave $\delta^{18}O$ excursions during HS2 may indicate a dating error in the WAIS Divide timescale. The remainder of our analysis is focused on comparison with the WAIS Divide $CH_4$ record which shares a common timescale (WD2014) with the composite $\Delta\varepsilon_{LAND}$ record. Absolute dating errors in WD2014 therefore do not hamper comparison of these two records.

Second, over the portion of the last glacial period covered by this record (~50-12 ka BP), $\Delta\varepsilon_{LAND}$ only consistently exceeds 0‰ (for more than a century) during the Younger Dryas and HS 1, 2, 4 and 5. $\Delta\varepsilon_{LAND}$ exhibits pronounced abrupt increases (downward in Figure 3) during HEs. Following HE 1, 2, 4 and 5 – all of predominant Hudson strait origin (Hemming, 2004) – $\Delta\varepsilon_{LAND}$ abruptly increases by ~0.1-0.3‰ over 200 to 300 years and remains generally high for ~1-2 thousand years. The timings of $\Delta\varepsilon_{LAND}$ maxima during each of HS 1, 2, 4, and 5 all occur within the proposed time windows of abrupt, large-scale

climate responses to each HE (Rhodes et al., 2015, Figure 4). Similarly, these proposed windows of HE climate correspond to periods of sustained above-average $\Delta\varepsilon_{LAND}$ values. Uncertainty in the measurement of $\delta^{18}O_{atm}$ and the computation of its time derivative preclude meaningful analysis of centennial-scale changes in $\Delta\varepsilon_{LAND}$ of below ~0.05‰.

Increases in $\Delta\varepsilon_{LAND}$ over HSs stand in contrast to repeated decreases during D-O warmings. Notably, atmospheric $CH_4$ increases during both D-O warmings and HE 1, 2, 4 and 5 (Northern Hemisphere cooling events). Thus, the relationship between $\Delta\varepsilon_{LAND}$ and atmospheric $CH_4$ reverses in sign between Northern Hemisphere warming and cooling. In Section 4 we

discuss the implications of this reversal for inference of the position of the thermal equator and tropical rainfall belts.

## 3.2 Seasonal cycle of terrestrial photosynthetic source water $\delta^{18}O$

We analyzed monthly-mean datasets of $\delta^{18}O_{precip}$ and GPP on land (Section 2.3) to test the sensitivity of GPP-weighted mean $\delta^{18}O_{precip}$ to the median latitude of oxygenesis over the seasonal cycle, which serves as a rough analogue for past shifts

in the thermal equator. Figure 5 shows December-January-February (DJF) and June-July-August (JJA) mean spatial distributions of $\delta^{18}O_{precip}$ and GPP. Throughout all seasons, $\delta^{18}O_{precip}$ generally trends isotopically lighter northward between the southern mid-latitudes and northern high-latitudes. The meridional trend in GPP, however, exhibits a strong seasonal dependence in its sign. As the northern terrestrial biosphere grows from boreal spring to summer, a northern maximum develops in the mid-to-high latitudes (Figure 6, Movie S2), shifting the centroid of terrestrial oxygen production northward.



During boreal fall and winter, northern hemisphere productivity decreases, shifting terrestrial oxygenesis southward toward the relatively seasonally invariable southern tropical maximum near 5°S.

For the purpose of quantifying meridional shifts in terrestrial GPP, we define the terrestrial oxygenesis equator (TOE) as the latitude of the terrestrial GPP centroid (Figure S1, Section 2.3). In simpler terms, half of all terrestrial oxygen production occurs to the north and half to the south of the seasonally varying TOE. The TOE shifts from ~5°S in boreal winter to ~35°N in boreal summer (Figure 7, Movie S1).

To determine the direct influence of $\delta^{18}O_{precip}$ on the $\delta^{18}O$ of $O_2$ produced by the terrestrial biosphere, we calculated monthly GPP-weighted mean $\delta^{18}O_{precip}$. Throughout the entire seasonal cycle, GPP-weighted mean $\delta^{18}O_{precip}$ varies by ~2.6‰, with the isotopically lightest and heaviest GPP-weighted precipitation occurring during boreal summer and winter, respectively. GPP-weighted mean $\delta^{18}O_{precip}$ was found to be strongly and significantly correlated with TOE latitude over the seasonal cycle ($r = 0.95$, $p < 0.0001$), with a mean sensitivity of -0.066‰ per degree TOE latitude. Applying a last glacial ice mask (Argus et al., 2014) such that GPP was set to zero of glaciated land areas at 26 ka, we find the mean sensitivity of GPP-weighted mean $\delta^{18}O_{precip}$ remains similar (-0.072‰/°latitude).

## 4 Discussion

Considering the observed seasonal-cycle relationship between the TOE and global terrestrial GPP-weighted mean $\delta^{18}O_{precip}$, here we propose a mechanism by which meridional shifts in the TOE are unambiguously recorded in the sign of $\Delta\varepsilon_{LAND}$ changes. Although the seasonal cycle is an imperfect analog for abrupt climate change during the last glacial period, the analysis presented in Section 3.2 highlights the importance of the meridional distribution of oxygen production for the GPP-weighted mean $\delta^{18}O$ of photosynthetic source water. Specifically, a northward shift of the TOE (e.g. during a D-O warming) independently lowers global GPP-weighted mean $\delta^{18}O_{precip}$ while a southward shift (e.g., during a HS) does the opposite. We suggest that the sign of this relationship is robust over long timescales although the magnitude likely varies. A simple experiment in which the modern terrestrial GPP climatology (with a 26-ka ice mask) is perturbed by reducing Northern Hemisphere GPP and increasing Southern Hemisphere GPP each uniformly by 25% suggests that several-degree-latitude shifts in the TOE may produce changes in terrestrial GPP-weighted mean $\delta^{18}O_{precip}$ that are comparable in magnitude (order 0.1‰)



to $\Delta\varepsilon_{LAND}$ changes over HSs (Figure S4). To further explore this notion, we consider our original definition of $\Delta\varepsilon_{LAND}$ (equation 1). In this definition, $\Delta\varepsilon_L$ encompasses all changes in global leaf water isotopic composition, which may be divided into separate source water and evapotranspiration components:

$$\Delta\varepsilon_L = \Delta\delta^{18}O_S + \Delta\varepsilon_{ET} \tag{6}$$

$\delta^{18}O_S$ is the change from present in oxygen production-weighted source-water $\delta^{18}O$ (having removed the influence of changes in seawater $\delta^{18}O$) and $\Delta\varepsilon_{ET}$ is the change from present in oxygen production-weighted evapotranspiration enrichment of leaf water $\delta^{18}O$.

Many studies of Dole effect variability over the last glacial cycle (e.g. *Bender et al.*, 1994; *Severinghaus et al.*, 2009; *Landais et al.*, 2010) have identified low-latitude precipitation as a key driver, based on coherence with orbital precession and atmospheric methane as well as high correlation with Chinese stalagmite $\delta^{18}O$. A recent quantitative, earth system model-based analysis of Dole effect variations over a simulated HS corroborates the hypothesized influence of low-latitude precipitation and finds that the increase in $\delta^{18}O_{atm}$ over a HS can be explained almost entirely by changes in terrestrial oxygen production-weighted $\delta^{18}O_{precip}$ (Reutenauer et al., 2015). *Reutenauer et al.* (2015) find that oxygen production-weighted relative humidity stays effectively constant between HS and background glacial conditions, which suggests negligible change in $\Delta\varepsilon_{ET}$ and implies (in terms of equation 6) that $\Delta\varepsilon_L \approx \Delta\delta^{18}O_S$

The latitude dependence of global GPP-weighted $\delta^{18}O_{precip}$ over the modern seasonal cycle implies that $\Delta\varepsilon_{LAND}$ should be sensitive to the position of the TOE over a HE At present, however, it is impossible to determine the exact TOE sensitivity of $\Delta\varepsilon_{LAND}$ because it is modulated by changes in the underlying spatial distribution of $\delta^{18}O_{precip}$. For instance, Chinese speleothem calcite records (Cheng et al., 2016; Wang et al., 2008, 2001) as well as isotope-enabled climate model simulations (Battisti et al., 2014; Liu et al., 2014) demonstrate that $\delta^{18}O_{precip}$ in southeastern China changes with EASM strength, such that local $\delta^{18}O_{precip}$ decreases during periods of an intense monsoon and increases with a weaker monsoon. Since this region is important for global oxygen production, it follows that a decrease in EASM intensity after a HE would act to increase $\Delta\varepsilon_{LAND}$. However, concurrent, smaller-amplitude decreases in southern tropical rainfall $\delta^{18}O_{precip}$ during a HS (Kanner et al., 2012), particularly in the productive Amazon lowlands (Wang et al., 2017), may somewhat offset this increase in $\Delta\varepsilon_{LAND}$. *Reutenauer et al.* (2015)





find that decreased southern tropical $\delta^{18}O_{precip}$ does not fully compensate for increased Northern Hemisphere $\delta^{18}O_{precip}$, and conclude that spatial changes in $\delta^{18}O_{precip}$ over the terrestrial biosphere impart a net increase on $\delta^{18}O_{atm}$ over a simulated HS.

Clearly, to understand $\Delta\delta^{18}O_S$ one must consider both 1) changes in regional $\delta^{18}O_{precip}$ ($\Delta\delta^{18}O_{precip}$), and 2) changes in the distribution of oxygen production (the TOE, to first order). We suggest that $\Delta\delta^{18}O_S$ is driven by the superposition of these two

components, both of which likely operate in the same direction. For a northward migration of the thermal equator and tropical rainfall belts, rainfall over productive northern regions like Southeast Asia becomes more isotopically depleted (Battisti et al., 2014; Cheng et al., 2016; Liu et al., 2014; Wang et al., 2008, 2001). Assuming Northern Hemisphere $\Delta\delta^{18}O_{precip}$ outweighs Southern Hemisphere $\Delta\delta^{18}O_{precip}$, a northward migration of the tropical rainfall belts imparts a negative signature on $\Delta\delta^{18}O_s$. Similarly, a northward shift of the thermal equator and therefore a warmer and wetter northern hemisphere (and cooler, drier

southern hemisphere) shifts the TOE northward as well. By increasing the relative contribution of the low-$\delta^{18}O_{precip}$ Northern tropics and mid-latitudes to global terrestrial $O_2$, global $\Delta\delta^{18}O_S$ will also decrease. Each of these effects on 1) and 2) – and especially their combination – would lead to a decrease in $\Delta\delta^{18}O_S$ and thus a decrease in $\Delta\varepsilon_{LAND}$ for an abrupt northern hemisphere warming (e.g. a D-O warming). In the opposite scenario (southward shift of thermal equator) during a HE, for example, the superposition of 1) and 2) would act to increase $\Delta\delta^{18}O_S$ because of a strong increase in Northern Hemisphere

$\delta^{18}O_{precip}$ and a southward shift of the TOE. Therefore, we suggest that the sign of a $\Delta\varepsilon_{LAND}$ change faithfully reflects the direction of meridional shifts in the thermal equator and tropical rainfall belts.

While changes in respiratory fractionation, $\Delta\varepsilon_{RL,}$ are poorly known, two leading suggestions for the response of $\Delta\varepsilon_{RL}$ to large-scale hydroclimate change each reinforce the notion that the sign of $\Delta\varepsilon_{LAND}$ change is a robust indicator of meridional shifts in tropical rainfall. First, *Reutenauer et al*. (2015) find that changes in terrestrial respiration over a modeled HS are

uncertain but small, one order of magnitude less important than GPP-weighted $\delta^{18}O_{precip}$ in explaining the modeled change in $\delta^{18}O_{atm}$. If true, this is consistent with the notion that $\Delta\varepsilon_{LAND}$ is largely controlled by $\Delta\varepsilon_L$, which mostly reflects $\Delta\delta^{18}O_S$. Second, if instead global $\Delta\varepsilon_{RL}$ does appreciably vary with shifts in tropical rainfall, a candidate mechanism is the "monsoon rectifier effect" which proposes that changes in $\Delta\varepsilon_{RL}$ may oppose and partially conteract changes in $\Delta\delta^{18}O_S$ at high latitudes but amplify changes in $\Delta\delta^{18}O_S$ in monsoon regions (Luz and Barkan, 2011; Severinghaus et al., 2009). This notion hinges on the close

coupling between local photosynthesis and respiration, along with the temperature dependence of soil respiration fractionation,



which is stronger (i.e. increased discrimination of $^{18}$O) in cold high-latitude soils (Angert et al., 2012). If, for example, northern high-latitude GPP were to increase (e.g. for a northward shift of the thermal equator during a D-O warming), so too would northern high-latitude soil respiration. In terms of equation 1, then, the lowering of $\Delta\varepsilon_L$ (due to contribution of the isotopically light high-latitude source water) would be somewhat offset by a more negative $\Delta\varepsilon_{RL}$ (stronger fractionation), leading to

attenuation of any change in $\Delta\varepsilon_{LAND}$. In monsoon regions, however, a stronger monsoon accompanying a northward shift of the thermal equator would result in both lower $\delta^{18}O_{precip}$ and weaker (less negative) soil respiration fractionation due to slow diffusion of oxygen in wet soils (Angert et al., 2001). In these regions, a lowering of $\Delta\varepsilon_L$ would be accompanied by less negative $\Delta\varepsilon_{RL}$, both of which act to lower $\Delta\varepsilon_{LAND}$. Therefore, in either case – whether $\Delta\varepsilon_{RL}$ is negligible or amplifies monsoon $\delta^{18}O_{precip}$ signals – one would still expect a net decrease (increase) in $\Delta\varepsilon_{LAND}$ for a northward (southward) shift of the thermal

equator and tropical rain belts.

The TOE-latitude dependence of past changes in $\Delta\varepsilon_{LAND}$ makes the composite SD-WD $\Delta\varepsilon_{LAND}$ record a useful complement to other records of past tropical hydroclimate change. A recent analysis of the high-resolution WD CH$_4$ record suggests that abrupt peaks in CH$_4$ during Hudson-strait HS (HS 1, 2, 4 and 5) are imprints of HE (Rhodes et al., 2015). Because past changes in tropical wetland CH$_4$ emissions are thought to be a bimodal function of the latitude of the thermal equator

(Rhodes et al., 2015), an abrupt northern hemisphere cooling and southward shift of the thermal equator and land-extension of the intertropical convergence zone (ITCZ) would intensify southern tropical precipitation and thus enhance CH$_4$ emissions. However, because northern hemisphere warming (e.g. D-O events) also increases atmospheric CH$_4$ emissions by stimulating northern wetland emissions, CH$_4$ measurements alone cannot definitively reveal the direction of meridional shifts in tropical rainfall. $\Delta\varepsilon_{LAND}$, which increases during and after CH$_4$ peaks representing HE (Figure 4), implies a southward shift of the TOE

and therefore provides strong evidence supporting the interpretation that these CH$_4$ peaks have a Southern Hemisphere source.

Future higher-resolution measurements of $\delta^{18}O_{atm}$ from WD alone may yield further insight into large-scale hydroclimate changes. A WD-only record would remove any uncertainty in $\Delta\varepsilon_{LAND}$ induced by the WD-SD synchronization process and allow for direct comparison with WD CH$_4$. Centennial-scale WD CH$_4$ variability may indicate important tropical dynamics (Rhodes et al., 2017) that could be elucidated by a companion high-resolution $\Delta\varepsilon_{LAND}$ record. For instance, the centennial-

scale oscillations in $\Delta\varepsilon_{LAND}$ following HE 1, which appear independently in both the SD and WD records, possibly hint that $\Delta\varepsilon_{LAND}$ may have spectral power in the centennial band.

Finally, we caution against over-interpretation of small $\Delta\varepsilon_{LAND}$ as clear indications of shifts in TOE latitude. Although global GPP-weighted mean $\delta^{18}O_{precip}$ is a first-order control on $\Delta\varepsilon_{LAND}$, small, under-constrained changes over time in

respiratory fractionation, evapotranspiration fractionation, the marine Dole effect, the terrestrial fraction of global oxygen production, and the slope of the TOE-$\delta^{18}O_{precip}$ relationship preclude meaningful analysis of 0.01‰-order $\Delta\varepsilon_{LAND}$ variability.

## 5 Conclusions

We analyzed a composite record of $\delta^{18}O_{atm}$ from the Siple Dome and WAIS Divide ice cores to compare synchronous instantaneous changes in the terrestrial fractionation of $\delta^{18}O_{atm}$ to records of Chinese speleothem $\delta^{18}O$ and WAIS Divide

atmospheric methane. Based on an analysis of the modern seasonal cycles of terrestrial oxygen production and production-weighted $\delta^{18}O_{precip}$, we propose a simple relationship between spatial shifts in terrestrial oxygenesis and $\delta^{18}O_{atm}$. Specifically, we identify a strong negative correlation between the centroid latitude of terrestrial oxygenesis and GPP-weighted mean $\delta^{18}O_{precip}$. This relationship suggests that positive $\delta^{18}O_{atm}$ (or $\Delta\varepsilon_{LAND}$) anomalies should accompany southward shifts of terrestrial oxygen production. Because productivity has strong hydroclimate controls, we propose that this mechanism may

explain much of the shared variability between past changes in the Dole Effect and proxies for low-latitude hydrological changes. Finally, positive excursions in $\Delta\varepsilon_{LAND}$ during HS 1, 2, 4 and 5 culminate in local $\Delta\varepsilon_{LAND}$ maxima which shortly follow small, abrupt increases in atmospheric methane. These changes in $\Delta\varepsilon_{LAND}$ imply southward shifts of terrestrial rainfall and therefore strongly support that interpretation that the near-contemporaneous spikes in methane – proposed to be evidence of a rapid atmospheric teleconnection during HE 1, 2, 4 and 5 – had a Southern Hemisphere source.

**Appendix A: Gas-loss correction and curve fitting**

Following *Severinghaus et al.* (2009), WD $\delta^{18}O$ measurements were corrected for enrichment due to gravitational settling and fractionation during gas loss. Gravitational settling leads to an increase in heavy to light gases, scaling nearly linearly with



their mass difference (Craig et al., 1988). Gravitational corrections were made to each measurement ($\delta^{18}O$, $\delta O_2/N_2$, $\delta^{15}N$, $\delta Ar/N_2$) as follows:

$$\delta_{gravcorr} = \delta_{meas} - \Delta m \cdot \delta^{15}N \tag{A1}$$

where $\delta_{meas}$ is a raw measured gas ratio, $\delta_{gravcorr}$ is its gravitationally corrected value, and $\Delta m$ is the mass difference between

heavy and light gases in amu (4 for $\delta O_2/N_2$, 12 for $\delta Ar/N_2$, and 2 for $\delta^{18}O$).

To determine $\delta^{18}O_{atm}$, $\delta_{gravcorr}$ was corrected empirically for fractionation due to gas loss. Slow loss of gases from the lattice

to microcracks causes mass-independent fractionation which affects $\delta O_2/N_2$ and $\delta Ar/N_2$, due to their different diameters, but

does not affect isotopes (Bender et al., 1995). Mass-dependent fractionation, which affects elemental and isotopic ratios, occurs

due to diffusive loss of gases through the microcracks (Kobashi et al., 2008). Following *Severinghaus et al.* (2009) we find a

plane of best fit (Figure S3) by regressing gravitationally corrected $\delta^{18}O$ replicate pair differences ($\Delta\delta^{18}O_{gravcorr}$) against

gravitationally corrected $\delta O_2/N_2$ and $\delta Ar/N_2$ pair differences ($\Delta\delta O_2/N_{2,gravcorr}$ and $\Delta\delta Ar/N_{2,gravcorr}$):

$$\Delta\delta^{18}O_{gravcorr} = A \cdot \Delta\delta O_2/N_{2,gravcorr} + B \cdot \Delta\delta Ar/N_{2,gravcorr} \tag{A2}$$

The regression coefficients governing this plane of best fit ($R^2=0.31$), *A* and *B,* are nearly equal and opposite in sign (A=-

0.00454; B=0.00447) similar to the finding of *Severinghaus et al* (2009) in SD samples, implying that fractionation of

$\delta^{18}O_{gravcorr}$ is indeed due to mass-dependent fractionation process (proportional to: $\delta Ar/N_{2,gravcorr}$ - $\delta O_2/N_{2,gravcorr}$), as the mass-

independent fractionation process equally fractionates $\delta Ar/N_2$ and $\delta O_2/N_2$. Finally, $\delta^{18}O_{atm}$ is calculated by removing this

empirically determined mass-dependent fractionation and fixing the mean $\delta^{18}O_{atm}$ value over the last 1000 years to 0‰ by

choosing *C* (below) to be 1.357:

$$\delta^{18}O_{atm} = \delta^{18}O_{gravcor} - A \cdot (\delta O_2/N_{2,gravcorr} + C) - B \cdot \delta Ar/N_{2,gravcorr} \tag{A3}$$

In order to reduce the influence of noise in calculating time derivatives of $\delta^{18}O_{atm}$ from SD, WD and the composite record,

each record was fit to a Fourier series by weighted linear least squares. Following *Severinghaus et al.* (2009), Fourier

amplitudes were assumed to be inversely proportional to their squared frequency (i.e. red spectrum). A Fourier series with

frequencies ranging from 1/697 to 3.2 cycles ka$^{-1}$ with spacing of 1/697 cycles ka$^{-1}$ was used for curve fitting. $\delta^{18}O_{atm}$ data

were weighted according to measurement uncertainties (pooled standard deviations of 0.0095‰ and 0.0085‰ for SD and WD,



respectively). SD outliers with anomalous $\delta^{15}N$ were downweighted as in *Severinghaus et al.* (2009). After determining Fourier

amplitudes, continuous fitted curves and analytical derivatives were evaluated at 10-year resolution.

**Appendix B: Details of Synchronization Method**

The gas timescale synchronization method implemented in this study (ALT, Fudge et al., 2014) optimizes error-weighted

tie point agreement and smoothness of the annual layer thickness profile. Note that because this study only concerns gas ages,

annual layer thicknesses are defined as vertical distances between two ice layers containing gas bubbles differing in age by

one year. Here we present: 1) the formally defined linear least squares problem and its uncertainty estimate, 2) methods for tie

point selection and uncertainty analysis, 3) an evaluation of the WD2014-synchronized SD gas age timescale, and 4) the results

of a sensitivity test using an alternate initial SD timescale for tie-point selection.

The generalized ALT cost function (equation 3) is given here in the full form used in our analysis (bold, capitalized and

lower-case letters denote matrices and vectors, respectively, while unbolded letters denote scalars):

$$\mathbf{C} = \mathbf{m^T G^T W G m} - 2\mathbf{m^T G^T W d} + \mathbf{d^T W d} + \alpha \mathbf{m^T L^T L m} \tag{B1}$$

where (as in equation 3) **G** maps annual layer thicknesses, **m**, to a depth-gas age relationship evaluated at tie point depths, **d**.

**L** is a second derivative operator, $\alpha$ is a trade-off parameter between smoothness and tie-point agreement, and **W** is a weighting

matrix with inverse tie point depth uncertainties ($1/\sigma^2$) along its diagonal. The time series of annual layer thicknesses that

minimizes equation B1, $\hat{\mathbf{m}}$, found at the zero crossing of $\frac{\partial \mathbf{C}}{\partial \mathbf{m}}$:

$$\hat{\mathbf{m}} = \mathbf{G^\# d} \tag{B2}$$

Equation B2 follows the common notation of *Lundin et al.* (2012) and *Aster et al.* (2013), such that $\mathbf{G^\#}$ is defined as:

$$\mathbf{G^\#} = (\mathbf{G^T W G} + \alpha \mathbf{L^T L})^{-1} \mathbf{G^T W} \tag{B3}$$

We note that an alternate formulation of $\mathbf{G^\#}$ is possible by using generalized singular value decomposition (Aster et al., 2013;

Lundin, 2012). We solved for $\hat{\mathbf{m}}$ via singular value decomposition and found that the maximum absolute disagreement of the

resulting annual layer thickness profile with the one calculated using B2 and B3 was negligible (order $10^{-8}$ meters).

The $1\sigma$ uncertainty of $\hat{\mathbf{m}}$ (for a fixed value of $\alpha$) is initially estimated by taking the square root of the diagonal values on

the covariance matrix, $\mathbf{V_m}$:



$$\mathbf{V_m = G^{\#}G^{\#T}} \tag{B4}$$

However, uncertainty in α also contributes to uncertainty in $\mathbf{\hat{m}}$. We account for this additional source of uncertainty by determining two candidate optimal values of α and evaluating resulting timescales using each value. These two candidate values of α minimize root mean square deviations between WD fitted-curve and SD discrete 1) $\delta^{18}O_{atm}$ values, or 2) $CH_4$

values, with SD gas ages shifted in each case to a new estimate of the WD2014-synchronized timescale. We evaluate $\mathbf{\hat{m}}$ separately for each value of α and take the absolute difference between these two time series of $\mathbf{\hat{m}}$ as an estimate of the 1σ uncertainty in our $\mathbf{\hat{m}}$ due to uncertainty in α. The final estimate of 1σ uncertainty in $\mathbf{\hat{m}}$ is then given by the quadrature sum of the covariance-matrix (fixed α) estimate (equation B3) and the estimated uncertainty due to α described above.

Tie points were estimated by matching abrupt transitions in $d\delta^{18}O_{atm}/dt$ and $CH_4$ in the WD and SD records. To find

$d\delta^{18}O_{atm}/dt$-based tie points, continuous time derivatives of the fitted $\delta^{18}O_{atm}$ curves were first evaluated as described in Appendix A. All local minima and maxima in $d\delta^{18}O_{atm}/dt$ were then identified and 36 abrupt transitions identifiable in each record were selected (each one between an identified local minimum and maximum). The midpoint time of a transition was found by determining when $d\delta^{18}O_{atm}/dt$ was halfway between the bounding extrema. The times at which $d\delta^{18}O_{atm}/dt$ was 25% and 75% between bounding extrema were taken as ±1σ uncertainty estimates. Figure B1 shows abrupt $d\delta^{18}O_{atm}/dt$ transition

midpoints, bounding extrema, and 25-75% markers identified in the SD (original timescale) and WD records. For each $d\delta^{18}O_{atm}/dt$ midpoint, corresponding SD depths and WD gas ages were used as tie points for the synchronization method.

A similar approach was employed to determine $CH_4$-based tie points using $CH_4$ records at SD and WD. Since WD $CH_4$ records (Rhodes et al., 2015; Sowers, 2012) are far higher resolution than SD $CH_4$ records (Ahn et al., 2012, 2014; Ahn and Brook, 2014; Brook et al., 2005; Ahn and Yang, unpublished), discrete SD $CH_4$ values were selected during abrupt transitions

for which the magnitude of SD and WD $CH_4$ values were in good agreement. 20 of these discrete mid-transition values were identified and used as tie-points. To find a WD gas age matching the SD depth of a discrete $CH_4$ measurement, the gas age of the nearest value in the 2-yr resolution WD record was determined. The 1σ uncertainty range for $CH_4$ tie points was estimated from the range of gas ages corresponding to all nearby WD $CH_4$ values that agreed with the selected SD $CH_4$ value to within ±15 ppb. Figure B2 illustrates this $CH_4$ tie point value matching and uncertainty process.





We also tested the sensitivity of the resulting timescale to the chosen $CH_4$ matching value range (±15 ppb) by varying it from ±5 to ±25 ppb to construct alternate timescales. Figure S5 shows deviations of these alternate timescales from the final timescale used in this study. The results of this test suggest that the choice of ±15 ppb for the final timescale is reasonable, because alternate timescales constructed with ranges of ±10 and ±20 ppb agree with the ±15 ppb-based timescale within ±1σ

uncertainty at all times, with a maximum absolute deviation of ~58 years.

Together, $d\delta^{18}O_{atm}/dt$ and $CH_4$ tie points were used to constrain equation B2 via the weighting matrix **W** to produce a best estimate of SD annual layer thicknesses at 10-year resolution from 50.12-0 ka BP (before 1950). Integrating this annual layer thickness time series yields a depth-gas age relationship. Figure 2 shows the annual layer thickness estimate, its uncertainty, and a comparison between our new ALT-based and the original *Brook et al.* (2005) SD timescales. To assess the validity of

the new SD gas timescale estimate, we calculated root mean squared deviations (RMSDs) between WD (fitted curve or interpolated) and SD (discrete) $\delta^{18}O_{atm}$ and $CH_4$ on the original and new timescales. Although a non-ideal evaluation metric for relatively flat portions of these records, RMSDs penalize timescale errors during transitions. The substantial RMSD reductions between the original and new timescale add confidence to the synchronization method. $CH_4$ RMSDs decrease from 33.7 to 18.3 ppb on the original and new timescales, respectively. $\delta^{18}O_{atm}$ RMSDs decrease from 0.029 to 0.022‰ on the

original and new timescales, respectively. Figures S5 and B3 show fitted and interpolated WD $\delta^{18}O_{atm}$ and $CH_4$ curves, respectively, alongside discrete SD $\delta^{18}O_{atm}$ and $CH_4$ measurements on the original and new timescales.

To investigate the sensitivity of our SD depth-gas age to the initial SD timescale, we repeated the SD tie point selection procedure and least squares minimization using an alternate SD timescale. This sensitivity test was motivated by concern that error in the original timescale would lead to error in the magnitude of the time derivative of $\delta^{18}O_{atm}$. The tie point method

implemented in this study (aligning the midpoints of $d\delta^{18}O_{atm}/dt$ transitions rather than value-matching) partially alleviates this concern. Nonetheless, for completeness, we carried out a sensitivity test using an alternate initial SD timescale for curve fitting, differentiation, and tie point determination to constrain synchronization method (equation 3). Because the *Brook et al.* (2005) timescale was constructed by tying SD gas measurements to the Meese/Sowers GISP2 (Greenland) ice core timescale (Meese, 1999), we added the difference in gas ages between the Meese/Sowers and more recent GICC05 timescales (Rasmussen et al.,

2014) to the *Brook et al.* (2005) SD timescale. This resulting alternate timescale differs from the original *Brook et al.* (2005)

timescale by up to ~2 ka in the interval from ~50 ka to present. This alternate timescale is notable older before ~45 ka, as

shown clearly by comparison of SD $d\delta^{18}O_{atm}/dt$ records on the original and alternate GISP2-tied timescales (Figure S6). The

close similarities in the magnitude of SD $d\delta^{18}O_{atm}/dt$ on the original and alternate timescales strongly indicates that the small

dating errors are not significant enough to substantially affect the magnitude of the $\delta^{18}O_{atm}$ derivative. Each of the 36 transitions

5    associated with $d\delta^{18}O_{atm}/dt$ tie points was identifiable on this alternate timescale (Figure S7). After determining the depths

corresponding to these midpoints on the alternate timescale (in order to determine tie point array **d** for equation B2) midpoint

depths were compared between the original and alternate timescales and found to agree within the 25-75% transition value

derived 2σ uncertainty range for 35 of 36 tie points. The same $CH_4$ tie point method was implemented and the resulting set of

46 depth-gas age tie points was used to estimate an alternate annual layer thickness profile via the ALT method (equation 3).

This alternate estimate was found to agree the original estimate within 2σ uncertainty throughout the entire time range

considered (Figure S8). We therefore conclude that given these relatively small dating errors of below ~2 ka, the $\delta^{18}O_{atm}$

derivative-based tie point determination method is justified.

## Acknowledgements

We thank Bruce Cornuelle for helpful discussions about curve fitting and linear optimization, Ross Beaudette for ice core

measurements, and the WAIS Divide and Siple Dome project members for making this work possible. WAIS Divide

atmospheric oxygen isotope measurements were made at Scripps Institution of Oceanography's Noble Gas Isotope Laboratory

and were supported by National Science Foundation grant 0538657.

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



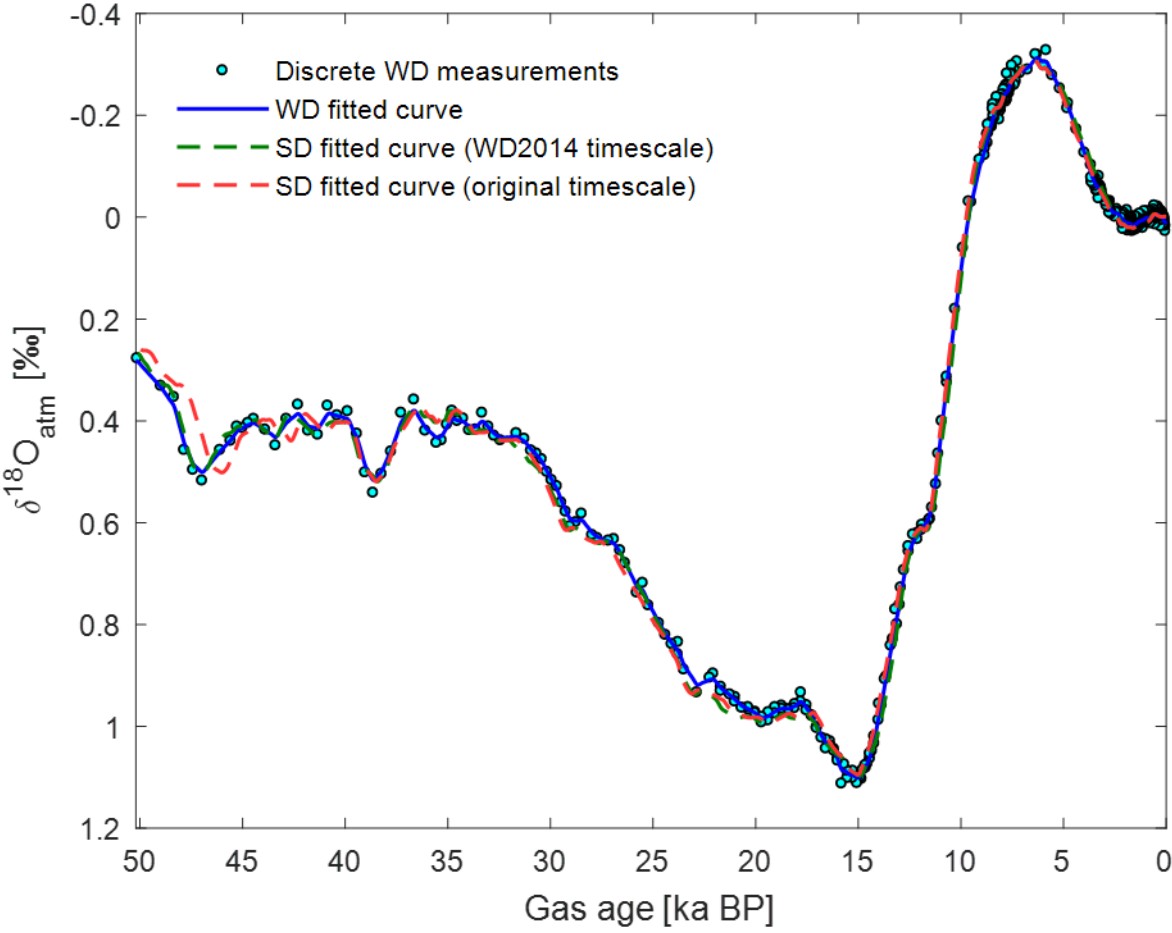

**Figure 1:** Discrete measurements from WAIS Divide (light blue markers) and fitted curves of $\delta^{18}O_{atm}$ from WAIS Divide (solid blue line) and Siple Dome on its original gas timescale (red dashed line, Brook et al., 2005) and WD2014-synchronized gas timescale (green dashed line, this study).





**Figure 2:** Annual layer thickness **(a)** from the original Siple Dome gas timescale (Brook et al., 2005) and WD2014-synchronized timescale (this study), and gas age differences between the original and WD2014-synchronized Siple Dome gas timescales **(b)** constrained by tie points selected from abrupt transitions in the CH$_4$ and d$\delta^{18}$O$_{atm}$/dt records at SD and WD.





**Figure 3:** Top: $\Delta\varepsilon_{LAND}$ (estimate of change in terrestrial fractionation of $\delta^{18}O_{atm}$ from modern value) calculated independently based on the fitted $\delta^{18}O_{atm}$ curves from WAIS Divide (blue) and Siple Dome (red, on WD2014-synchronized timescale) as well as the composite WD-SD record (black); Middle: unsmoothed and smoothed (200-yr boxcar) records of composite calcite $\delta^{18}O$ from Southeast Asian speleothems (Cheng et al., 2016) corrected for changes in seawater $\delta^{18}O$ (Waelbroeck et al., 2002); Bottom: WAIS Divide $CH_4$ measurements (Rhodes et al., 2015; Sowers, 2012). Shaded regions indicate periods of climate impact due to Heinrich Events 1, 2, 4, and 5 identified by *Rhodes et al.* (2015), coincident with the highest observed glacial values of $\Delta\varepsilon_{LAND}$.





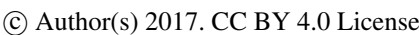

**Figure 4:** Atmospheric $CH_4$ measured at WAIS Divide and composite WD-SD $\Delta\varepsilon_{LAND}$ throughout and following Heinrich Events 1, 2, 4 and 5. $\Delta\varepsilon_{LAND}$ is highlighted green during periods of climate impact from Heinrich Events 1, 2, 4, and 5 proposed by *Rhodes et al.* (2015), beginning with an abrupt $CH_4$ rise (blue dashed line with shaded $\pm1\sigma$ dating uncertainty) and ending with another $CH_4$ rise of a presumed northern source (red dashed line with shaded $\pm1\sigma$ dating uncertainty).





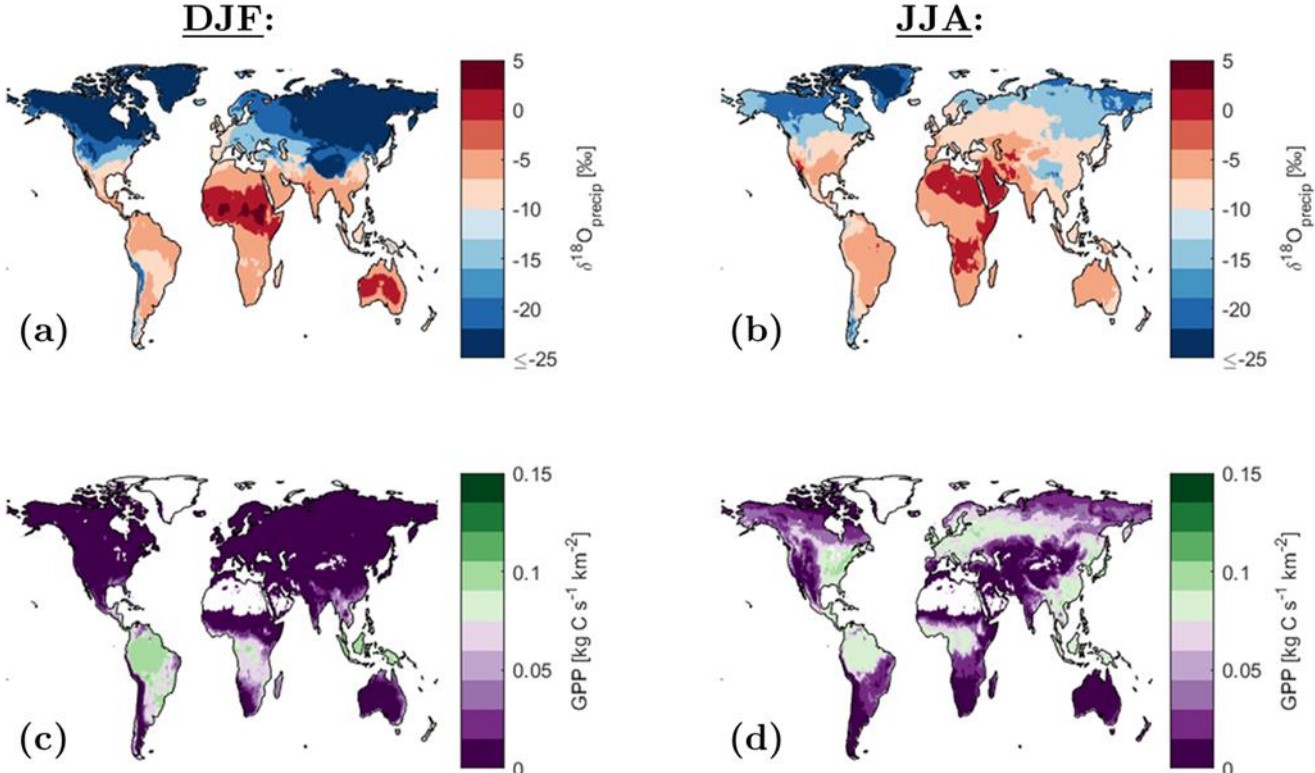

**Figure 5:** Mean terrestrial December-January-February (**a, c**) and June-July-August (**b, d**) $\delta^{18}O_{precip}$ (**a, b**) and GPP (**c, d**). GPP data used in this study are monthly values averaged across a 30-year (1982-2011) dataset of upscaled eddy covariance observations (Jung et al., 2011). Monthly $\delta^{18}O_{precip}$ isoscapes are derived from long-term (1960-2009) monthly GNIP (Global Network of Isotopes in Precipitation) data (Terzer et al., 2013). Data gaps (e.g. over oceans, Greenland, Sahara Desert) are left unfilled in this plot.

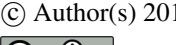



**Figure 6:** Zonally integrated terrestrial gross primary productivity as a function of latitude for monthly-mean data averaged over years from 1982 to 2011 (Jung et al., 2011). The dashed red line shows monthly-mean TOE latitude.



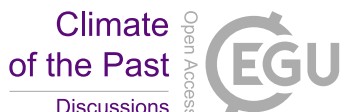

**Figure 7:** Mean monthly terrestrial $\delta^{18}O_{precip}$, weighted by oxygen production (GPP), is significantly correlated to the latitude of terrestrial oxygenesis equator ($r = 0.95$, $p < 0.0001$). Green (June-July-August) and red (December-January-February) boxes highlight extreme seasonal shifts in TOE latitude and terrestrial GPP-weighted $\delta^{18}O_{precip}$. Associated text describes the simple mechanism suggested by this study in which meridional shifts in the thermal equator affect $\delta^{18}O_{atm}$ and $\Delta\epsilon_{LAND}$ by shifting terrestrial oxygenesis to latitudes of higher or lower $\delta^{18}O_{precip}$.







**Figure B1:** Overview of tie point selection process from $d\delta^{18}O_{atm}/dt$ records at WD (blue) and SD (red, on original *Brook et al*., 2005 gas timescale). Green circles (N=36) indicate midpoints of abrupt $d\delta^{18}O_{atm}/dt$ transitions recorded in both records. Smaller yellow markers indicate 25% and 75% values between local extrema (black dots) bounding each abrupt transition. ±1σ tie point uncertainty was taken from the range of 25-75% markers.





**Figure B2:** Overview of tie point selection process from CH₄ records at WD (blue) and SD (red, on original *Brook et al.*, 2005 gas timescale). Green circles (N=20) indicate selected discrete SD CH₄ values and nearest matching WD values. Smaller yellow markers indicate range of WD values agreeing with each selected SD CH₄ value within ±15ppb, taken as an estimate of ±1σ tie point uncertainty.



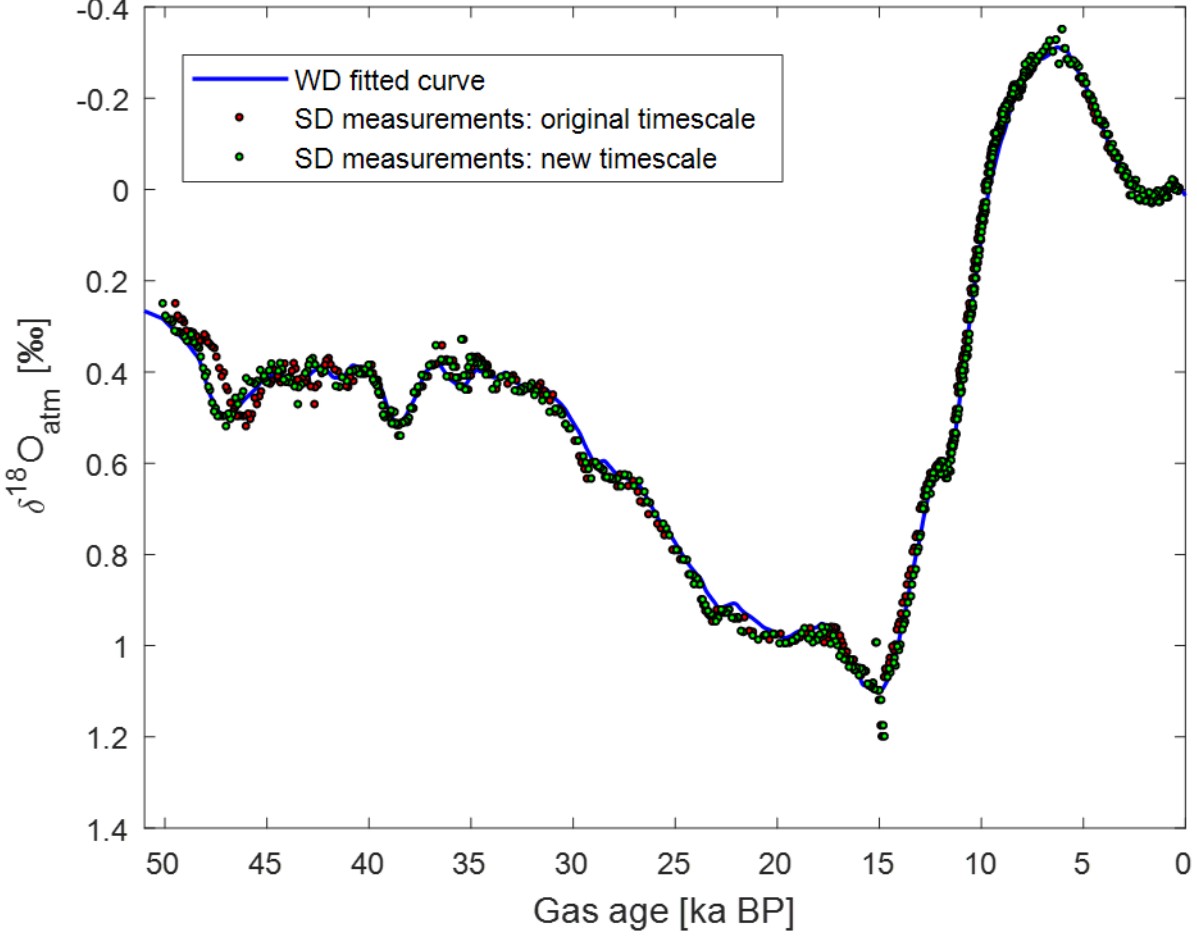

**Figure B3:** $\delta^{18}O_{atm}$ in SD and WD: discrete SD measurements on original gas timescale (blue markers, Brook et al., 2005) and on new gas timescale (green markers, this study) are superimposed on WD fitted curve (blue line) on WD2014 gas timescale (Buizert et al., 2015). Root mean squared deviations between WD fitted curve and SD discrete measurements are 0.029‰ on 5   the original timescale and 0.022‰ on the new timescale.





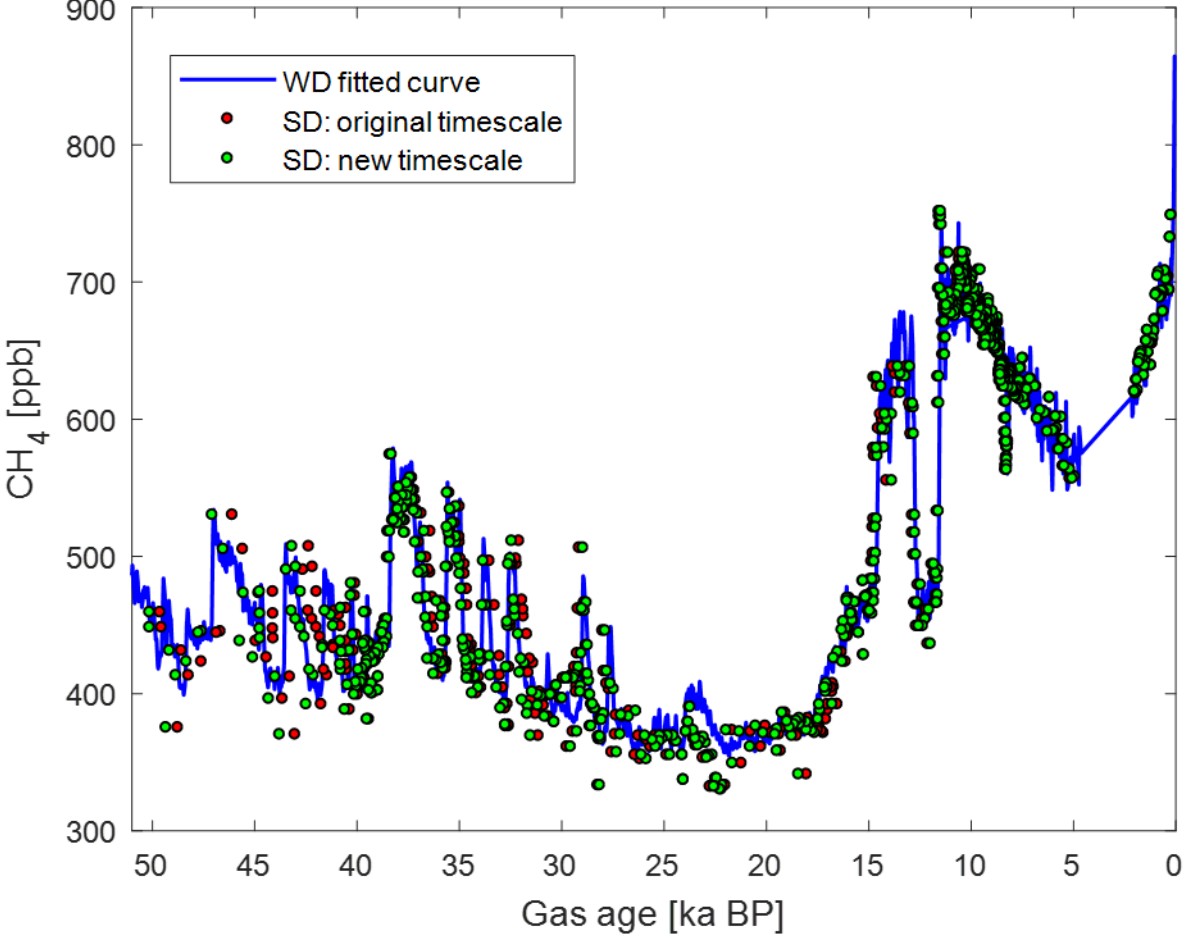

**Figure B4:** CH₄ in SD (Ahn et al., 2012, 2014; Ahn and Brook, 2014; Brook et al., 2005; Ahn and Yang, unpublished) and WD (Rhodes et al., 2015; Sowers, 2012): discrete SD measurements on original gas timescale (blue markers, Brook et al., 2005) and on new gas timescale (green markers, this study) are superimposed on WD interpolated curve (blue line) on WD2014 gas timescale (Buizert et al., 2015). Root mean squared deviations between WD fitted curve and SD discrete measurements are 33.7 ppb on the original timescale and 18.3 ppb on the new timescale.