# Peer review of "Does $\delta^{18}$ O of O2 record meridional shifts in tropical rainfall?"

_Climate of the Past, 2017_

## Referee Comment (RC1) · Anonymous Referee #1 · 26 Jul 2017

Seltzer et al. measured $\delta$18O of occluded air in an ice core drilled on the West Antarctic Ice Sheet Divide (WD). The reported record spans the past 50 ka and is of extreme high quality worthy of publication by its own right. The exceptional precision and high temporal resolution make it possible to de-convolute the integrated record of atmospheric signal and calculate a record of variations in terrestrial fractionation. The new WD record almost perfectly correlates with the previously published one for the Siple Dome ice core, thus demonstrating the robustness of both. The above was possible by meticulous care given to all the difficult mass spectrometric measurements. As well, the reported results were carefully corrected for isotope effects of gas loss, gravity and thermal diffusion by using the ratios of nitrogen isotopes, and the ratios of oxygen and argon to nitrogen. There is no question that the substantial data set in the manuscript

is of superb quality. In a further step the records were carefully dated by numerous control points of known ages. This made it possible to relate the atmospheric records of methane and terrestrial fractionation to the records of inferred strength of the SW Asian Monsoon and Heinrich Events (Fig. 3).

In addition, the authors have made an analysis of present day controls over the isotopic composition of air oxygen. They are aware of the complex nature of the combined biospheric and hydro-climate system. They have derived a plausible scheme showing the response of atmospheric composition to seasonal shifts in rain belts and then used it for interpreting the record of terrestrial fractionation. It shows that southern shifts in the terrestrial oxygenesis equator, and perhaps weakening of respiratory fractionation in tropical soils, match quite well the strength of the SW Asian monsoon. They also discuss the correspondence of terrestrial fractionation and atmospheric methane and the underlying controlling factors. This discussion however, lacks clarity and so are the derived conclusions. For example, in the abstract they state that maxima in terrestrial fractionation are synchronous with methane peaks. This is opposite to what they show in Fig. 3 where methane peaks lag behind terrestrial fractionation maxima. This requires clarification perhaps by stating that the maxima in terrestrial fractionation correspond to secondary methane rises. Regardless of the clarification, it appears that terrestrial fractionation is a better recorder of low latitude variations in rain. In conclusion, I am impressed by the high quality of the research described in the manuscript and recommend its publication after the discussion is revised. In addition, I suggest rewriting the abstract and introduction using simpler and less technical terminology thus making them more appealing to broader audience.

---

## Referee Comment (RC2) · Anonymous Referee #2 · 31 Jul 2017

It has long been suspected that changes in d18Oatm and the Dole effect are dominated by changes in the hydrological cycle in low latitudes. With better analyses methods it surfaced that also the long lived d18Oatm signal has a millennial scale component. Supported by recent complex modeling efforts (Reutenauer et al., 2015) Seltzer et al., qualitatively explain the observed d18Oatm variations with changes of the GPP weighted precipitation signature in the tropics. They provide a thorough and very complete analysis of the individual components of the oxygen cycle. Although the idea of tropical hydrology dominating the d18Oatm signal is not new their analysis convincingly demonstrates that shift in the tropical GPP weighted precipitation pattern can explain the observations.

Minor comments: Introduction: Please explain in 2-3 sentences what DO events are.

[Figure]

Page 3, lines 19-20: This statement is wrong. The analysis in Bender et al., 1994 is based on the Vostok CH4 data available at that time. It simply did not have the resolution to detect millennial variability and neither did the d18Oatm record. Please rephrase to make clear that the statement in Bender et al. is of a more general nature. Page 6, line 15: add MODERN mean monthly gridded datasets Page 6, line 24: I do not understand the meaning of this sentence; "...GPP refers to an emission, rather than a flux...". My understanding is that equation 4 is a GPP weighted d18O precipitation signature and equation 5 is the GPP weighted location of this mean precipitation. Page 9, line 20: "independently" of what? Page 10, line 14: Briefly explain the importance of relative humidity in terrestrial oxygen production. Page 10, line 20: add (East Asian Summer Monsoon) after EASM

---

## Author Comment (AC1) · 25 Aug 2017

We thank the reviewer for his or her positive feedback and constructive advice. Based on these comments, we plan to improve the clarity of our introduction, abstract and discussion sections by using simpler terminology and being more specific where needed. In particular, we realize that the comparison of local maxima in CH$_4$ and $\Delta\epsilon_{LAND}$ could be confusing as presently worded. We thank the reviewer for bringing this to our attention. In its present form, the abstract states that "maxima in $\Delta\epsilon_{LAND}$ are synchronous with or shortly follow WD CH$_4$ peaks assumed to mark abrupt climate responses to Heinrich events." In our revised paper, we will clarify in the abstract and main text that these "WD CH$_4$ peaks" refer to small amplitude (tens of ppb) local maxima in CH$_4$ within Heinrich Stadials 1,2, 4 and 5. They do not refer to the much larger amplitude (100s of

ppb) millennial-scale local maxima associated with Dansgaard-Oeschger events.

---

## Author Comment (AC2) · 25 Aug 2017

We thank the reviewer for his or her helpful comments and positive review. Below are our specific responses/plans to revise according to this reviewer's feedback:

-Introduction: we plan to provide more background on DO and Heinrich Events in our revised introduction. We appreciate the suggestion.

-Page 3, lines 19-20: we will revise this sentence to more accurately reflect scope of Bender et al., 1994 by re-wording "Millennial-scale variability in the Dole Effect" to "Variability in the Dole Effect throughout the past glacial-interglacial cycle"

-Page 6, line 15: we will add "modern" here and thank the reviewer for pointing out this potential source of confusion.

-Page 6, line 24: the preceding sentence (lines 22-24) explains that GPP data (originally fluxes given in carbon mass per grid cell area) have been converted to total emissions in order to remove their latitude-dependence so that total oxygen production can be compared at different latitudes. The reviewer is correct in stating that equation 4 refers to terrestrial GPP-weighted mean $\delta^{18}O_p$ and equation 5 refers to the latitude at which half of all terrestrial GPP is produced to the north and half to the south.
-Page 9, line 20: we agree that the word "independently" is confusing here, and we will remove it in the revised paper. The intention was to point out that, independent of any change in $\delta^{18}O_p$, GPP-weighted mean $\delta^{18}O_p$ will change due to shifts in the location of oxygen production. This point is more clearly made later in the discussion.
-Page 10, line 20: we will define EASM in parentheses as suggested.

———————————————————

---

## Author Response (AR1)

**List of changes:**

*\*\*changes in direct response to comments of **reviewer #1** are listed in **green** and those in direct response to **reviewer #2** are listed in **blue**\*\**

- Abstract: discussion of Heinrich-related $CH_4$ rises clarified
- Introduction: more background provided on D-O events; more precise summary of glacial-interglacial variability in Dole Effect
- Methods: clarified that seasonal GPP and $\delta^{18}O_{precip}$ datasets are modern observations
- Discussion: made minor wording changes for clarity suggested by reviewer #2 and listed in response to reviewer #2 (below); provided brief explanation for evapotranspiration-induced $\delta^{18}O$ enrichment of leaf water
- Conclusion: again clarified Heinrich-related $CH_4$ rises per suggestion of reviewer #1
- References: added appropriate reference for evapotranspiration fractionation of leaf water $\delta^{18}O$
- Data availability: added section with DOI link to published dataset

**Point-by-point responses to reviewer comments:**

*\*\*our responses to direct comments are interspersed (in **bold font**) in each reviewer's comments below \*\**

*Reviewer #1:*

*Seltzer et al. measured δ18O of occluded air in an ice core drilled on the West Antarctic Ice Sheet Divide (WD). The reported record spans the past 50 ka and is of extreme high quality worthy of publication by its own right. The exceptional precision and high temporal resolution make it possible to de-convolute the integrated record of atmospheric signal and calculate a record of variations in terrestrial fractionation. The new WD record almost perfectly correlates with the previously published one for the Siple Dome ice core, thus demonstrating the robustness of both. The above was possible by meticulous care given to all the difficult mass spectrometric measurements. As well, the reported results were carefully corrected for isotope effects of gas loss, gravity and thermal diffusion by using the ratios of nitrogen isotopes, and the ratios of oxygen and argon to nitrogen. There is no question that the substantial data set in the manuscript is of superb quality. In a further step the records were carefully dated by numerous control points of known ages. This made it possible to relate the atmospheric records of methane and terrestrial fractionation to the records of inferred strength of the SW Asian Monsoon and Heinrich Events (Fig. 3). In addition, the authors have made an analysis of present day controls over the isotopic composition of air oxygen. They are aware of the complex nature of the combined biospheric and hydro-climate system. They have derived a plausible scheme showing the response of atmospheric composition to seasonal shifts in rain belts and then used it for interpreting the record of terrestrial fractionation. It shows that southern shifts in the terrestrial oxygenesis equator, and perhaps weakening of respiratory fractionation in tropical soils, match*

*quite well the strength of the SW Asian monsoon. They also discuss the correspondence of terrestrial fractionation and atmospheric methane and the underlying controlling factors. This discussion however, lacks clarity and so are the derived conclusions. For example, in the abstract they state that maxima in terrestrial fractionation are synchronous with methane peaks. This is opposite to what they show in Fig. 3 where methane peaks lag behind terrestrial fractionation maxima. This requires clarification perhaps by stating that the maxima in terrestrial fractionation correspond to secondary methane rises.*

**Our response: Changes (underlined in marked manuscript below) were made to both the abstract and discussion section to clarify that the abrupt rises in atmospheric methane which correspond to local $\Delta\varepsilon_{LAND}$ maxima occur within Heinrich stadials and are on the order of several tens of ppb. These small-amplitude $CH_4$ excursions are thought to indicate an abrupt response to Heinrich events (Rhodes et al., 2015).**

*Regardless of the clarification, it appears that terrestrial fractionation is a better recorder of low latitude variations in rain. In conclusion, I am impressed by the high quality of the research described in the manuscript and recommend its publication after the discussion is revised. In addition, I suggest rewriting the abstract and introduction using simpler and less technical terminology thus making them more appealing to broader audience.*

**Our response: In our introduction we have now provided more background information on millennial scale climate variability during the last period, expanding on Dansgaard-Oeschger events in particular. We have also revised the abstract as mentioned in the previous comment.**

*Reviewer #2:*

*It has long been suspected that changes in d18Oatm and the Dole effect are dominated by changes in the hydrological cycle in low latitudes. With better analyses methods it surfaced that also the long lived d18Oatm signal has a millennial scale component. Supported by recent complex modeling efforts (Reutenauer et al., 2015) Seltzer et al., qualitatively explain the observed d18Oatm variations with changes of the GPP weighted precipitation signature in the tropics. They provide a thorough and very complete analysis of the individual components of the oxygen cycle. Although the idea of tropical hydrology dominating the d18Oatm signal is not new their analysis convincingly demonstrates that shift in the tropical GPP weighted precipitation pattern can explain the observations.*

*Minor comments:*

*Introduction: Please explain in 2-3 sentences what DO events are.*

**Our response: We have added an overview of D-O events to the introduction.**

*Page 3, lines 19-20: This statement is wrong. The analysis in Bender et al., 1994 is based on the Vostok CH4 data available at that time. It simply did not have the resolution to detect millennial*

*variability and neither did the d18Oatm record. Please rephrase to make clear that the statement in Bender et al. is of a more general nature.*

**Our response: We have changed "Millennial-scale variability in the Dole Effect" to "Variability in the Dole Effect throughout the past glacial-interglacial cycle"**

*Page 6, line 15: add MODERN mean monthly gridded datasets*

**Our response: We have added "modern" here in the revised manuscript.**

*Page 6, line 24: I do not understand the meaning of this sentence; ". . .GPP refers to an emission, rather than a flux. . .". My understanding is that equation 4 is a GPP weighted d18O precipitation signature and equation 5 is the GPP weighted location of this mean precipitation.*

**Our response: (the following explanation is repeated here from our online response: The preceding sentence (lines 22-24) explains that GPP data (originally fluxes given in carbon mass per grid cell area) have been converted to total emissions in order to remove their latitude-dependence so that total oxygen production can be compared at different latitudes. The reviewer is correct in stating that equation 4 refers to terrestrial GPP-weighted mean d18Op and equation 5 refers to the latitude at which half of all terrestrial GPP is produced to the north and half to the south.)**

*Page 9, line 20: "independently" of what?*

**Our response: We have removed "independently" here in the revised manuscript. (Here is our response to the comment as given online in the interactive discussion: We agree that the word "independently" is confusing here, and we will remove it in the revised paper. The intention was to point out that, *independent* of any change in d18Op, GPP-weighted mean d18Op will change due to shifts in the location of oxygen production. This point is more clearly made later in the discussion.)**

*Page 10, line 14: Briefly explain the importance of relative humidity in terrestrial oxygen production. Page 10, line 20: add (East Asian Summer Monsoon) after EASM.*

**Our response: We have made this change in the revised manuscript.**

[revised manuscript text omitted]